# Natural Tropical Oscillations phase impact on stationary and westward travelling planetary waves

Kseniia A. Didenko<sup>1,2</sup>, Andrey V. Koval<sup>1,3</sup>, Tatiana S. Ermakova<sup>1,3</sup>, Aleksey S. Fadeev<sup>1</sup>, Luyang Xu<sup>4</sup>, Ke Wei<sup>5</sup>

<sup>1</sup>Department of Atmospheric Physics, Saint Petersburg State University, St. Petersburg, 199034, Russia

Correspondence to: Kseniia A. Didenko (didenko@izmiran.ru), Ke Wei (weike@mail.iap.ac.cn)

Abstract. We performed a series of numerical experiments to study the main patterns of the Quasi-Biennial Oscillation phase of the zonal wind in the equatorial stratosphere (QBO) and the El Niño-Southern Oscillation (ENSO) influence on stationary and westward travelling atmospheric planetary waves (PWs) with different zonal wave numbers and periods. The simulation was carried out for boreal winter conditions using the Middle and Upper Atmosphere Model (MUAM). The results showed that the joint effect of the considered tropical oscillations can significantly up to tens of percent change the amplitudes of the PW in the areas of their maxima. Under the El Niño regardless of the QBO phase the amplitude maxima of the stationary PW with wave number 1 (SPW1) shift toward high latitudes. The amplitude structure of SPW2 is basically opposite to the SPW1 structure. Increases of the upward wave activity fluxes of quasi- 5-, 10-, 7-day PWs, as well as the amplitudes of 10- and 7-day PWs are modelled when easterly QBO phase is superimposed on El Niño phase. Conversely, attenuations of the individual PW amplitudes and wave activity fluxes are typically observed under the westerly QBO, as well as under La-Niña/westerly QBO conditions combination in special cases, such as SPW1. The PW study is important due to their significant influence on the middle and upper atmosphere circulation, including the configuration of the stratospheric polar vortex whose deformation can influence the occurrence of extreme weather events, in particular, in the Arctic and Asia-Pacific region during the boreal winter.

### 1 Introduction

Recently, there have been an increasing number of studies that have underscored the role of long-range teleconnections in the climate variations as well as the formation of various extreme winter/summer events (Ding and Wang, 2005; Wang et al., 2020; Rudeva and Simmonds, 2021). This includes the interactions between atmospheric layers and interconnection of low-and high-latitude regions associated with natural tropical oscillations impacting both the atmospheric temperature and wind

<sup>&</sup>lt;sup>2</sup>Space weather prediction center, Pushkov Institute of Terrestrial Magnetism, Ionosphere, and Radio Wave Propagation (IZMIRAN), Troitsk, Moscow, 108840, Russia

<sup>&</sup>lt;sup>3</sup>Department of Meteorological Forecasts, Russian State Hydrometeorological University, St. Petersburg, 195196, Russia

<sup>&</sup>lt;sup>4</sup>Beijing Weather Forecast Center, Beijing China, 100089

Of Sciences System Research, Institute of Atmospheric Physics, Chinese Academy of Sciences, Beijing, 100029, China

patterns, as well as the development of the winter dynamics across the Northern Hemisphere (Garfinkel et al., 2018; Baldwin et al., 2019; Rao et al., 2019a; Liu et al., 2024; Zhang et al., 2024). Under the influence of these teleconnections, variability of East Asian winter/summer monsoons, polar vortex, frequency and strength of sudden stratospheric warming (SSW) events, cold waves, upper-tropospheric jet streams etc., are observed leading to extreme weather conditions (Chen et al., 2005; Garfinkel et al., 2012; Huang et al., 2021; Kumar et al., 2022).

Among the oscillations in low latitudes the most notable are the quasi-biennial oscillation of zonal wind (QBO) and the El Niño-Southern Oscillation (ENSO). There are two main phases of the QBO – easterly and westerly characterized by the direction of the zonal wind in the stratosphere of equatorial latitudes with an average phase change period of 28 months. We will also extract two main phases For ENSO – cold (La Niña) and warm (El Niño), determined by temperature anomalies in the eastern or central equatorial region of the Pacific Ocean. These phenomena not only specify the wind, temperature and cloud regime, but are also mainly controlled by wave acceleration (Halpert and Ropelewski, 1992; White et al., 2015). The signal from QBO and ENSO is also transferred to the polar latitudes by atmospheric waves of a global scale (planetary waves, PWs) (Garfinkel and Hartmann, 2008; Calvo et al. 2009; Lee and Kyung Jin, 2024; Koval et al., 2022b; Ermakova et al., 2019).

Planetary atmospheric waves are generated as a result of perturbation of atmospheric parameters that exhibit a periodic structure in the zonal direction. Koval et al. (2023) demonstrated that the propagation of individual PW can alter the speed of the background wind and the components of the residual meridional circulation up to 5%, especially in the areas of their maxima. The capacity of PWs to propagate not only horizontally, but also vertically provides an effective mechanism for the transfer of energy and momentum from the troposphere up to thermospheric heights, as well as the exchange of wave energy between the hemispheres (Holton and Tan, 1980; Holton et al. 1995; Koval et al., 2022a; 2025). Waves interact with the mean flow and with each other as they propagate upwards causing anomalies in atmospheric circulation. A particularly striking example is the occurrence of sudden stratospheric warmings (SSW) in the winter polar stratosphere under the influence of waves (Matsuno, 1971; Baldwin et al., 2021; Pogoreltsev et al., 2014; 2015). The stratospheric polar vortex may shift southward, split into two or stretch out during SSW events. This can lead to the alterations to the jet stream's location, consequently affecting the storm trajectories, the formation of cold waves over East Asia, North America, and Europe, and the regional modulation of winter temperature and wind patterns, as well as the subsequent summer season particularly when combined with tropical oscillations (Thompson et al., 2002; Yang et al., 2002; Lü et al., 2020; Zhang et a al., 2022; Yan et al., 2023). For example, Zhang et al. (2022) proposed a mechanism explaining the influence of major SSW, accompanied by the displacement of the stratospheric polar vortex on the unprecedented cold wave that occurred in East Asia in January 2021. An analysis of cold air outbreaks in Canada and midwestern USA after January, 2019 SSW in the presence of lower Barents-Kara Sea ice based on both observations and model experiments is presented in Zhang et al. (2020). Lü et al. (2020) showed that anomalous Siberian snow accumulation could have played an important role in the 2018 SSW occurrence. This SSW with a vortex split type and predominant planetary waves of zonal wave 2 regime has led to cold extremes over Europe. In turn, Lu et al. (2021) discussed the favourable conditions that contributed to the development of the SSW in January 2021, which included the amplification of planetary wavenumbers 1 and 2, and these also serve as a mechanism for transmitting signals from ENSO. Chen et al. (2005) also noted the dominant contribution of the zonal wavenumber-2 pattern of planetary waves to variability of the East Asian winter monsoon.

Studies have been conducted to investigate the joint effect of OBO and ENSO in various aspects. For instance, in the works of Garfinkel and Hartmann (2007), Kumar et al. (2022), and Liang et al. (2023) variations in stratospheric temperatures, mass transport, and composition distribution, along with mean meridional circulation, polar vortex, and subtropical westerly jet stream, were investigated. Specifically, Kumar et al. (2022) showed the influence of QBO on the polar vortex, where westerly OBO corresponds to a colder and stronger vortex, while easterly OBO is linked to a warmer and weaker vortex. However, the OBO's effect on strengthening extratropical patterns was more pronounced during the La Niña phase, and the anomalies associated with OBO become more intricate during El Niño. Rao et al. (2019b) reached similar conclusions, indicating that the favourable conditions for the SSW event that affect the strength of polar vortex include the easterly OBO. moderate El Niño, and solar minimum. Conversely, Garfinkel and Hartmann (2007) demonstrated in an earlier work that the uncontaminated signals from OBO and ENSO are comparable in magnitude, and if one factors causes the vortex to weaken, the effect of the other factor within the sample under consideration is diminished. Despite the growing number of studies on the joint effect of OBO and ENSO on global atmospheric circulation, a detailed examination of their joint effect on the structures of global atmospheric waves has yet to be conducted, and the significance of such research is unquestionable. The global circulation of the middle and upper atmosphere is predominantly wave-driven; specifically, PWs generate a meridional circulation motion, as described by Haynes et al. (1991), in accordance with the "downward control principle". Consequently, alterations in planetary wave activity have a substantial impact on the temperature regime along the PW waveguides (e.g., Koval et al., 2019; 2023; 2025).

The purpose of this study is to investigate the influence of diverse QBO and ENSO phases on wave processes and their structural characteristics during boreal winter. A general circulation model, the Middle and Upper Atmosphere Model (MUAM) was utilized to facilitate the examination of such influences. MUAM allows taking into account various boundary and background conditions in modelling the general circulation of the atmosphere. Numerical modelling allowed us to carry out idealized experiment, without imposition of other atmospheric processes that could introduce errors, which are not possible using reanalysis measurements. Based on the MUAM simulation results, this study focused on both stationary planetary waves (SPWs) and westward travelling PWs with periods of 4-10 days.

## 2 Methods and approaches

75

## 2.1 MUAM characteristics, configuration and the processes considered

The general atmospheric circulation was simulated using a three-dimensional nonlinear model of the middle and upper atmosphere MUAM (Fröhlich et al., 2003; Pogoreltsev et al., 2007) in this study. As a result of the modelling, it was possible to obtain distributions of meteorological parameters from the level of 1000 hPa to the heights of the ionospheric

layer F2 or approximately 300 km with 56 height levels. The horizontal resolution is  $5.625^{\circ} \times 5^{\circ}$  in longitude and latitude, respectively. MUAM solves a standard system of primitive equations adapted for a spherical coordinate system. The MUAM radiation unit takes into account the heating of the atmosphere in the ultraviolet and visible regions of the spectrum, as well as cooling in the infrared bands. Ion friction, molecular and turbulent viscosity, thermal conductivity, and 3-dimensional ozone distribution are taken into account. The main parameters calculated by the model are the zonal, meridional and vertical components of wind speed, geopotential, and temperature. The main characteristics and processes that are taken into account during the MUAM modelling are described in detail in Jacobi et al. (2017); Ermakova et al. (2019); Medvedeva et al. (2019) and references therein, and a description of numerical experiments of the current version of the MUAM is presented in Koval et al. (2021).

The main advantage of the model is the ability to take into account different phases of tropical oscillations when modelling the general atmospheric circulation. For example, a nudge is used to account for the QBO in the MUAM, i.e. zonally averaged zonal wind fields are stepwise pushed to observations. Since the model is not able to self-consistently reproduce QBO in the stratosphere, the background and initial distributions of zonal wind fields, which are considered for years with different phases of QBO, are set in the model (Koval et al., 2022a; 2022b). These distributions are calculated from data sets consistent with a certain phase of the QBO. Two data sets of 10 years each were obtained using the method of decomposing meteorological fields into empirical orthogonal functions (EOF) for the typical easterly (1987, 1989, 1996, 1998, 2000), 2003, 2005, 2007, 2010, and 2012) and westerly (1983, 1985, 1993, 1995, 1997, 1999, 2002, 2004, 2008, and 2013) phases of the QBO. For more information about the method, see Friedrich et al. (1993) and Wallace et al. (1993). Description of this method adaptation to the MUAM calculations can be found in Koval et al. (2022b).

In this work one of the latest versions of the MUAM was used, including parameterizations of atmospheric heating rates caused by the release of latent heat, which take into account both diurnal and longitude oscillations and dependence on the El Niño-Southern oscillation phase ENSO (Ermakova et al., 2019). Years corresponding to the warm El Niño phase and the cold La Niña phase were selected based on a multivariate index – MEI (Wolter and Timlin, 2011). Using the table of available MEI values, 5 years were selected for the conditions of the each of two ENSO phases; 1983, 1992, 1998, 2003, 2010 for El Niño conditions and 1989, 1999, 2000, 2008, 2011 for La Niña conditions. Temperature and geopotential fields from these two data sets were averaged and used as initial conditions in the MUAM.

The scheme of numerical experiments consists of several stages of setup. At the first stage the MUAM simulation begins with an initial windless atmosphere with a climatological averaged global vertical temperature profile. During the first 30 days calculation the gravity waves parameterizations are not taken into account in the model, and the height of the geopotential at the lower boundary does not change. The longitude variations of the geopotential, which are the sources of stationary planetary waves, are set at the next stage. During the first 120-140 days, the MUAM uses average daily heating rates. After a day in the range of 120-140 days daily variations in heating and an additional predictive equation for the geopotential at the lower boundary as an source of PWs implicates. Starting from the 300 model day seasonal changes in the Sun's zenith angle are launched, and days 300-390 correspond to December-February.

To achieve the statistical significance of the results obtained, ensemble calculations are necessary. In this work 4 ensembles containing of 10 solutions ("runs") each were obtained, corresponding to 4 combinations of QBO and ENSO phases: El Niño/eQBO (easterly phase of QBO), El Niño/wQBO (westerly phase of QBO), La Niña/eQBO and La Niña/wQBO. The model ensembles of calculations are formed by taking into account the various phases of the mean wind and planetary waves vacillations in the middle atmosphere, which in turn are set by the date of inclusion of daily variations in solar heating and generation of normal atmospheric modes. The initial and background conditions, with the exception of the QBO and ENSO conditions, are the same in all model calculations. The monthly mean PW amplitudes, the intensity of the mean zonal flow and temperature can vary significantly from one model run to another. This variability is interpreted as interannual (Pogoreltsev, 2007; Pogoreltsev et al., 2019).

#### 2.2 Planetary waves in MUAM

The SPWs, quasi-5-day PW, and quasi-10-day PW have been extensively studied since 1980-s (Salby 1981, 1984; Hirooka and Hirota 1985; Hirooka 1986, 2000; Madden 2007; Sassi et al. 2012). The quasi-10-day westward propagating planetary wave probably is less-known in comparison with quasi-5-day PW but its amplitude and wave activity analysis are still carried out in connection with SSW possible development and ionospheric disturbances (Wang et al., 2021), and polar mesospheric clouds formation (Su et al, 2024). Thus, the obtained results described below are an important addition to those already published. The quasi-4-day PW is getting investigated in regard to the link with SSW (Sassi et al. 2012; Ma et al. 2020; Yamazaki et al, 2021). The quasi-7-day wave is essentially less studied; however, the importance of this wave mostly due to its nonlinear interactions with other waves. Thus, the obtained results that concern quasi-4-day and quasi-7-day waves can be understood as initial steps in assessment of these waves' contribution to wave activity flux.

An important advantage of the MUAM is the reproduction fidelity of the structures of planetary atmospheric waves – stationary planetary waves (SPW) and atmospheric normal modes (NMs). The reliability of the simulated PW's structures and magnitudes is confirmed by comparison with the results of processing reanalysis data, satellite and radar data presented in Garcia et al. (2004); Gavrilov et al. (2018); Forbes and Zhang (2015); Mukhtarov et al. (2010); Riggin et al. (2006); McDonald et al. (2011); Li et al. (2021), which was discussed in many studies (e.g., Koval et al., 2018; 2019; 2022a; 2023). The accuracy and temporal variability of the PWs simulated by the MUAM has recently been investigated by comparing simulated PW wavelet spectra with observations (Didenko et al., 2024). From amplitude wavelet spectra of westward travelling PWs with zonal wave numbers 1 and 2 it was shown that MUAM reproduces interannual variability observed in reanalysis data. Besides 4-, 5-, and 16-day PWs amplitude maximum and PWs nonlinear interactions simulated by MUAM were founded in EOS MLS (The Earth Observing System Microwave Limb Sounder, Waters et al. (2006)) data.

The sources of PWs at the lower boundary are set on the basis of geopotential heights in the atmosphere lower layers for SPW and additional terms of the thermal balance equation in the form of time-dependent sinusoidal harmonics with different zonal wave numbers for westward travelling NMs (Pogoreltsev et al., 2007). In this work, SPWs with zonal wave numbers m = 1 and m = 2 were studied, as well as westward travelling NMs with periods of about 5-, 10-days with m = 1 and 4-,7-days

- with m = 2, i.e., first symmetric and antisymmetric modes according to the classification proposed by Longuet-Higgins (1968). The periods of travelling PWs correspond to the resonant periods of the atmosphere reaction to disturbances at the lower boundary caused by baroclinic/barotropic instability. A description of the specifying the latitudinal structure of NMs by Hough functions is presented in Swarztrauber and Kasahara, 1985 and its application to the results of modeling with the MUAM was described by Pogoreltsev (1999); Koval et al. (2022a); Didenko et al. (2024).
- Waveguides regions with a positive value of quasi-geostrophic zonal-mean refractivity index squared,  $n^2$  (Matsuno, 1970) were calculated to analyze the background conditions for propagation of PWs. This index is calculated using the following formula (Albers et al., 2013):

$$n^{2}(\varphi, z) = \frac{\overline{q_{\varphi}}}{u - c} - \left(\frac{m}{a \cos \varphi}\right)^{2} - \left(\frac{f}{2NH}\right)^{2},\tag{1}$$

where  $\varphi$  is latitude, u is zonal wind component, c is PW zonal phase velocity, a is the Earth's radius, f is the Coriolis parameter, N is the Brunt-Vaisala frequency, and H is the atmospheric pressure scale height. Subscripts denote partial derivatives.  $\overline{q_{\varphi}}$  is the latitudinal gradient of zonal-mean potential vorticity, calculated as follows:

$$\overline{q_{\varphi}} = \frac{2\Omega\cos\varphi}{a} - \frac{1}{a^2} \left(\frac{(\overline{u}\cos\varphi)_{\varphi}}{\cos\varphi}\right)_{\varrho_0} - \frac{f^2}{\rho} \left(\rho \frac{\overline{u}_z}{N^2}\right)_z, \tag{2}$$

where  $\rho = \rho_0 \exp(-z/H)$  is the standard density in log-pressure coordinates, with the sea-level reference density  $\rho_0$ .

The waves influence on the mean flow and vice versa, i.e. the wave-mean flow momentum exchange, under conditions of various combinations of QBO and ENSO can be estimated using the Eliassen-Palm flux (EP flux). The meridional and vertical components of the EP flux,  $F_m = (F_m^{(\varphi)}, F_m^{(z)})$ , can be calculated in order to quantify changes in circulation and the possible contribution of PW to this change:

$$F_m^{(\varphi)} = \cos \varphi \left( \bar{u}_z \frac{\overline{v'\theta'}}{\overline{\theta_z}} - \overline{u'v'} \right), \tag{3}$$

$$F_m^{(z)} = \cos \varphi \left( \left( f - \frac{(\overline{u}\cos\varphi)_{\varphi}}{a\cos\varphi} \right) \frac{\overline{v'\theta'}}{\overline{\theta_z}} - \overline{w'v'} \right). \tag{4}$$

In Eq. 3 and 4 the overlines and primes indicate the mean zonal value and deviation from this value, respectively, v and w are the meridional and vertical components of the wind;  $\theta$  is the potential temperature.

#### 3 Results and discussion

190

The fields of hydrometeorological parameters modelled by the MUAM were used to study the structures and wave activity fluxes of the SPW and westward travelling NMs under the various conditions related to the different phase combinations of tropical oscillations. To assess the statistical significance of the results obtained, the two-month time interval under study (January-February) was divided into four 15-day intervals. The amplitudes and phases of the PW in geopotential height were

calculated for all 40 model runs and for each of four subintervals using the Fourier transform and the least square fitting method. Similar approach was used when studying the influence of solar activity (Koval, 2019; Koval et al., 2019; 2025) and OBO (Koval et al., 2022a) on PW structure up to the thermosphere.

The calculation results of the studied SPW1 and SPW2 (stationary planetary waves with zonal wave numbers of 1 and 2, respectively), as well as the westward travelling PW with a zonal wave number m=1, having periods of 5- and 10-days; with a zonal wave number m=2, periods of 4- and 7-days for January-February are shown in Fig. 1-6. Structurally, these figures look as follows: in the panels **a**) the values of the amplitudes of the corresponding PW (colour shading) are shown; negative refractive index (areas beyond the waveguide) are hatched with gray; and the vectors of the EP fluxes (arrows) averaged over all 40 runs are shown. I.e. panels **a**) show model-mean climatic values. Panels **b**) show deviations of the amplitudes (colour shading) and of the EP flux vectors (arrows) of the corresponding PW in the combination of El Niño/eQBO from the model-mean climatic value, and hatched areas show statistically insignificant increments (at 95%). Panels **c**), **d**) and **e**) show the same as b), but under the combinations of El Niño/wQBO; La Niña/eQBO and La Niña/wQBO, respectively.

The waveguides in panels a) of Fig. 1-6 are shown between hatched areas and correspond to regions with a positive refractive index  $n^2$ , showing the regions of PW propagation. For all deviations from the model mean climatic values 95% statistical significance was obtained using the Student's paired t-criterion, calculated from 40 paired values (10 model runs × 4 time subintervals). The method used here to assess statistical significance is described in detail in Koyal (2019).

## 3.1 Stationary planetary waves

The results for SPW1 and SPW2 in Fig. 1a and 2a show that the maximum amplitudes of these waves are modelled in the mid- and high-latitude upper stratosphere and lower mesosphere of the Northern Hemisphere, which is typical for the boreal winter conditions. SPW2 is characterized by another amplitude maximum near the level of 30 km, 60° N. The SPW wave activity fluxes are directed along the waveguides upwards and towards low latitudes – area outside the gray hatching and arrows in Fig. 1a and 2a. SPW2 wave activity flux can even penetrate into low latitudes – arrows in Fig. 2a. An increase in EP fluxes and, as a result, in the SPW's amplitudes is also observed in Fig. 1a and 2a in the lower thermosphere (80-110 km for SPW1 and 80-90 km for SPW2) around 60° N. The reasons for the observed PW amplitude increases related to secondary waves generation are presented in Laštovicka (2006); Hoffmann et al. (2012); Xu et al. (2012); Didenko et al. (2024); Koval et al. (2024). These studies discuss the observed and modelled increases in the PW amplitudes above the waveguide interruption region, which are caused by PW modulation, for example, by gravity waves, solar atmospheric tides and/or nonlinear interactions between PWs.

The most interesting are the increments of the SPW amplitudes and EP fluxes under conditions of various combinations of ENSO and QBO, presented in panels b) – e). Under El Niño conditions, regardless of the QBO phase, the maximum of the amplitudes of SPW1 shifts to high latitudes around the 40 km level (in this area, the amplitude increases by 35% compared to the simulated mean climatic ones). This is accompanied by an increase in the upward wave activity flux – Fig. 1b and 1c

and creates favourable conditions for the SSW formation and therefore affects the strength of the stratospheric polar vortex. As shown in Lifar et al. (2024) when considering the same ensemble simulations, under combinations of El Niño/eQBO and El Niño/wQBO SSW event occurred in 9 out of 10 and 8 out of 10 members of the ensemble, respectively, including the major SSW leading to the stratosphere polar vortex disruption. Under the conditions of La Niña/eQBO, an increase in the amplitudes of SPW1 is observed in the region of its climatic maxima and reaches 20%, as shown in Fig. 1d. Significant SPW1 attenuations corresponding to decreases in EP fluxes are modelled under the conditions of La Niña/wQBO (Fig. 1e). In the area of SPW1 climatic maximum the amplitude decreases by 20%, and at the level of 40 km of middle latitudes by a factor of 4. Under these conditions, the MUAM does not reproduce SSW (Lifar et al., 2024).

Figure 1. a): amplitude of the geopotential height variations caused by SPW1 (gp.m., shaded), EP flux (m²/s², arrows). Hatched areas show negative refractive index beyond the waveguide. Data is averaged over January–February and all MUAM simulations; b-e): respective amplitude and EP flux increments due to change of ENSO/QBO phase, hatched areas show insignificant data at 95%. Vertical EP flux component is multiplied by 200.

The behaviour of the SPW2 amplitudes in the stratosphere under El Niño conditions (Fig. 2b and 2c) is basically opposite of the behaviour of SPW1 amplitudes, i.e., there is a weakening of the amplitudes in the regions of climatic maxima and this

weakening can reach 10%. The largest increases in SPW2 amplitudes by 25% and the increments of the upward EP flux are modelled under La Niña conditions: at high latitudes of the Northern Hemisphere under eQBO in Fig. 2d and at low latitudes of the Northern Hemisphere under wQBO in Fig. 2e.

Our calculations also showed that statistically significant increments of SPW's amplitudes are also observed in the thermosphere, above 120 km. The areas of significant increments for SPW1 are seen under wQBO conditions and they are accompanied by corresponding changes in the SPW1 wave activity. The variability of SPW2 amplitudes in the thermosphere is observed under the conditions of all combinations: a decrease in amplitudes under El Niño conditions and an increase under La Niña conditions at high latitudes of the Northern Hemisphere. Statistically significant variability of the SPW2 amplitudes is also observed in the thermosphere of the middle latitudes, with the greatest variations under La Niña/wQBO conditions. The mechanisms of influence of stratospheric oscillations on the propagation of PWs into the thermosphere and, accordingly, a detailed analysis of the effects in the thermosphere remain beyond the scope of this article and will be studied the future.

Figure 2: The same as Fig. 1 but for the SPW2.

#### 255 3.2 Westward travelling normal atmospheric modes

As shown in Fig. 3-6 (panels a)) westward travelling NMs with different periods have amplitude maxima in both Southern and Northern Hemispheres, while some of them are an effective mechanism of wave energy transmission between the hemispheres. This is evidenced by the directions (mainly horizontal) and magnitudes of the EP wave activity fluxes in the low-latitude stratosphere shown in Fig. 4a, 5a and 6a.

The 5-day NM with m=1 has maximum amplitudes in the mid-latitude mesosphere and the lower thermosphere of the Southern Hemisphere (Fig. 3a), which is consistent with the observational data (Didenko et al., 2024). In addition, significant amplitudes of this wave are modelled in the equatorial and mid-latitude lower thermosphere and the mid-latitude stratosphere. Apart from these, under various combinations of ENSO and QBO conditions in the region of the climatic maximum the 5-day NM amplitude varies slightly. The amplitude attenuation by ≈10% is observed under eQBO at the middle latitudes above 100 km in the Southern Hemisphere and at 50 km in the Northern Hemisphere - Fig. 3b and 3d. The 265 5-day NM amplitude increases by 10% under the wQBO in the same regions - Fig. 3c, 3e. An increase in the horizontal transfer of wave activity flux from the Southern Hemisphere to the Northern Hemisphere is observed at eOBO, and vice versa at wQBO, regardless of the ENSO phase at a level of about 80-120 km. An increment/decrement of the upward EP flux in the high-latitude southern thermosphere is modelled under a combination of El Niño/eOBO and El Niño/wOBO, 270 respectively. Fig. 3e shows an increase of 5-day PW amplitude during La Niña/wQBO in the area of a significant SPW weakening in the northern stratosphere (Fig. 1e). The assessment of the OBO effect only when comparing the middle and right panels of Fig.3 shows the weakening of 5-day PW during eQBO (middle panels) versus wQBO (right panels) up to 40 km in the Southern Hemisphere and up to 60 km in the Northern Hemisphere, which corresponds to our previous estimates presented in Koval et al., (2022a).

Figure 3: The same as Fig. 1 but for the westward travelling atmospheric NM with  $\tau = 5$  days, m = 1.

The structure of the travelling 10-day NM with m = 1 differs from that of 5-day NM, which is explained by its lower phase velocity and, in accordance with the formula (1), a smaller waveguide width compared to the 5-day PW (gray areas in Fig. 3a and 4a). The peaks of its amplitudes are observed from the middle to high latitudes of the Northern Hemisphere at levels of 40-80 km. In the middle latitudes of the lower thermosphere of both hemispheres, small amplitudes of the considered PW are also modelled – Fig. 4a. A statistically significant increase in the 10-day NM amplitudes by 20% is modelled under the eQBO conditions: at El Niño in the lower thermosphere (Fig. 4b) and at La Niña in the northern mid-latitude stratosphere (Fig 4d). An increase in the upward wave activity flux in the Northern Hemisphere corresponds to the positive amplitude changes: at levels of 40-120 km under the conditions of El Niño/eQBO in Fig. 4b and below 80 km under La-Niña/eQBO in Fig. 4d. The situation is reversed under the wQBO conditions, where PW weakening correlates with a decrease of the EP flux in the mesosphere/lower thermosphere of the middle latitudes at El Niño and in the high ones at La Niña.

Figure 4: The same as Fig. 1 but for the westward travelling atmospheric NM with  $\tau = 10$  days, m = 1.

The amplitude distribution similar to 10-day NM is typical for 7-day NM with m=2, but with smaller amplitudes – Fig. 5a. A statistically significant increase in the amplitudes of 7-day PW in the areas of its maxima and upward EP flux increments are seen under El Niño/eQBO conditions. At the same time, the anomaly can reach 30% in some areas. These positive anomalies are balanced by negative ones under the rest of ENSO/QBO combinations. The greatest decrease of the 7-day NM amplitudes in the regions of its maxima and downward EP flux increments are modeled under the La Niña/wQBO conditions. The interhemispheric transfer of wave energy from the winter hemisphere to the summer hemisphere above 70 km is the strongest under El Niño/eQBO and the weakest under La Niña/wQBO conditions. Below this level in the low-latitude stratosphere, increase in southward EP flux and PW amplitudes are registered during La Niña phase for both QBO (Figs. 5d, 5e). Opposite tendency is shown under El-Niño/wQBO in Fig. 5c.

Figure 5: The same as Fig. 1 but for the westward travelling atmospheric NM with  $\tau = 7$  days, m = 2.

The variability of 4-day NM with m=2 under conditions of various ENSO and QBO phases combinations is shown in Fig. 6. The largest amplitudes of the considered PW are observed in the mid-latitude stratosphere of the Southern Hemisphere. In addition, the maxima of the 4-day PW are also located in the lower mid-latitude thermosphere of both hemispheres and in the middle latitudes of the Northern Hemisphere at levels of 30-60 km – Fig. 6a. The weakening of this wave above 40 km in the Southern Hemisphere and above 60 km in the Northern Hemisphere is associated with the narrowing of the waveguide and its interruption in the middle and high latitudes of both hemispheres at the corresponding altitudes (gray areas in Fig. 6a). Notable statistically significant increases in the 4-day PW amplitudes (accompanied by an increase in the upward EP flux) are modeled in the stratosphere of the Southern Hemisphere at eQBO and decreases (accompanied by a weakening of the EP flux) – under wQBO, regardless of the ENSO phase. This is driven by an increase in interhemispheric wave energy transport in the stratosphere. There is also an interesting effect of wave activity spreading from the northern stratosphere to the southern one, while the wave amplitude in the Southern Hemisphere is greater than in the northern one. This effect, visible in Fig. 6a, is enhanced by the eQBO in Fig. 6b and 6d. The weakening of the 4-day PW in the stratosphere is

accompanied by a weakening of the indicated meridional transfer of wave activity. The results in Fig. 5 and 6 show the low statistical significance of the increments of PW amplitudes with m=2 due to their high variability.

Figure 6: The same as Fig. 1 but for the westward travelling atmospheric NM with  $\tau = 4$  days, m = 2.

Special attention in the study of atmospheric waves and wave processes is now being paid to the MLT (mesosphere-lower thermosphere) region. As can be seen from panels a) of Fig. 1-6 (gray areas), interruptions of PW waveguides are mainly observed in this region. Despite this, in the MLT area waves still dominate wind and temperature regimes, and consequently, the study of their parameters in this layer is necessary to analyze the influence of PW sources and PW propagation in the lower and upper atmospheric layers (Hagan et al., 2009; Funke et al., 2010; Vincent, 2015).

The results of modeling under conditions of various QBO and ENSO combinations for the MLT region show similar statistically significant increases of the amplitudes of SPW1 and SPW2 under El Niño phase, regardless of the QBO phase – Fig. 1b, 1c, 2b and 2c. SPW amplitudes and wave activity fluxes under La Niña conditions vary differently, leading to more substantial contribution of the QBO phase to these changes – Fig. 1d, 1e, 2d and 2e. The variability of westward travelling PW in the MLT region is much more complex and is influenced by both ENSO and QBO phases. For example, the 5-day NM amplitudes variability in the Southern Hemisphere is mainly due to the ENSO phase, and in the Northern Hemisphere

this variability is individual in each combination – Fig. 3. In turn, in structurally similar 10- (m = 1) and 7- (m = 2) day NM, there are no statistically significant considerable amplitude increments at La Niña/eQBO – Fig. 4d and 5d. Amplitudes increase under El Niño/eQBO, – Fig. 4b and 5b. The attenuation of these PW amplitudes in the MLT region is determined by the QBO phase and is observed under its westerly phase – Fig. 4c, 4e, 5c and 5e. The statistically significant changes in the 4-day PW amplitudes is subject to the QBO phase, i.e., the amplitude increases under the eQBO and decreases under the wOBO.

## 4 Conclusion

This study examines the variability of atmospheric waves of a global scale under the various combinations of long-period tropical oscillation phases, such as the Quasi-Biennial Oscillation (QBO) of the zonal wind in the equatorial stratosphere and the El Niño-Southern Oscillation (ENSO). The study was conducted using the results of numerical modelling with the Middle and Upper Atmosphere Model (MUAM). Numerical experiments were carried out for the winter conditions of the Northern Hemisphere. The analysis was focused on the amplitudes of stationary planetary waves with zonal wave numbers m = 1 and m = 2, westward travelling 5-, 10-day atmospheric normal modes (NMs) with m = 1, and 4-, 7-day NMs with m = 1, along with their wave activity fluxes.

The results of the numerical experiments revealed a significant difference in the considered PW's structures under various combinations of QBO and ENSO. In particular, it was shown that:

- Under El Niño conditions, regardless of the QBO phase, the maximum of SPW1 amplitudes shifts 20 km downwards and towards high latitudes. The amplitudes increase by 35% in comparison with the simulated climatic ones, which is accompanied by an increase in the upward wave activity fluxes. Significant attenuation of SPW1 is modelled under La Niña/wQBO conditions. In the northern stratosphere the increment sign of SPW2 amplitudes is basically opposite to the increment sign of SPW1 amplitudes under El Niño conditions. These changes in SPW structures are reflected in changes in the corresponding EP-fluxes.
- Similar features in the SPW1 and SPW2 amplitude increments at El Niño are modelled in the MLT region, regardless of the QBO phase. Increases of the amplitudes of SPW1 and SPW2 are shown under El Niño phase.
- The 5-day NM amplitude in the region of its climatic maximum varies slightly under various combinations of QBO and ENSO phases. An increase in the horizontal transfer of 5-day PW activity flux from the Southern Hemisphere to the Northern Hemisphere at the heights of the lower thermosphere is observed at eQBO, while a decrease is observed at wQBO, irrespective of the ENSO phase.
- A 20% increase in the 10-day PW amplitudes is modelled under the eQBO conditions. The situation is opposite under wQBO. An increase in the upward wave activity flux in the Northern Hemisphere high latitudes at levels of 80-120 km is observed only under El Niño/eQBO conditions.
  - The most pronounced statistically significant increase in the 7-day PW amplitudes in the areas of their maxima and corresponding EP flux changes is modelled under El Niño/eQBO conditions. Overall, this strengthening in

- different regions of the Northern Hemisphere is counterbalanced by a weakening of the 7-day NM for other phase combinations.
  - A significant enhancement of the 4-day PW, accompanied by an increase in the upward EP flux, is observed in
    the Southern stratosphere at eQBO, while a weakening, accompanied by a decrease in the EP flux, is observed
    at wQBO, regardless of the ENSO phase.
- The variability of westward travelling NMs in the MLT region is influenced by both QBO and ENSO phases: the ENSO phase causes remarkable variations in the 5-day PW amplitudes in the Southern Hemisphere. An increase in the amplitudes of 10- and 7-day PWs is observed at El Niño/eQBO, and the weakening is mainly associated with the westerly QBO phase. QBO also determines the change in the 4-day PW amplitudes.

The results confirmed the existing views that natural tropical oscillations, originating in low latitudes, significantly affect the structure of planetary waves and their wave activity fluxes, not only in the regions of their climatic maxima but also throughout the middle atmosphere and thermosphere of both hemispheres. Concurrently, in contrast to numerous other studies that assess the influence of QBO and ENSO on atmospheric waves separately, this study shows that the maximum statistically significant increases of PW amplitudes and wave activity fluxes occur under specific combinations of QBO and ENSO phases. Given the large effect of ENSO cycle on tropical convection, it is reasonable to suppose that ENSO may have a role in modulating QBO behaviour. On average, according to Taguchi (2010), the QBO signals may exhibit faster phase propagation during El Niño than during La Niña conditions, and the amplitude of the QBO is weaker during El Niño, which is discussed in detail in (Kawatani et al., 2019).

When interpreting our results, we should take into account the inevitable limitations imposed by the use of a relatively simple mechanistic model of the MUAM. In particular, the model lacks interactive photochemistry, does not model cloudiness, and does not take into account interaction with the ocean. However, when considering the main trends in large-scale dynamic processes, these limitations appear to be advantages: taking into account changes in only long-term tropical oscillations in the modeling, we consider their effects in their pure form, without the imposition of other processes. At the same time, as mentioned in Section 2, the ability of the MUAM model to realistically reproduce wave atmospheric dynamics has been repeatedly confirmed by research.

Further investigation into the structures, characteristics and propagation of PWs under the development of various oscillations is of great practical importance for describing the formation mechanisms of atmospheric circulation anomalies, thereby facilitating a better understanding of variability in tropospheric processes related to temperature and wind anomalies, particularly in the Asia-Pacific Region.

**Data availability.** In accordance with the statement 1296 of the Civil Code of the Russian Federation, Russian State Hydrometeorological University (RSHU) has all rights on the MUAM code. A permission for computer code usage access is needed for a reader from the RSHU Rector at the address 79, Voronezhskaya street, St. Petersburg, Russia, 192007, phone: 007 (812) 372-50-92. It is possible to obtain such permission with the assistance of the authors. All presented patterns in the

- paper are archiving to Zenodo. Graphical information in this study is obtained using Grid Analysis and Display System (GrADS) that is a free software developed by to the NASA Advanced Information Systems Research Program.
- Author contribution. All authors have made valuable contributions to the writing and editing of the paper, data analysis and imaging of the results. KAD: conceptualisation, numerical modelling, writing the final version of the paper; AVK: conceptualization, numerical modelling, data processing and statistical processing; TSE: data processing, consulting, English editing; ASF: data processing, numerical simulations; LX and KW: consulting and proofreading.
  - **Competing interests.** The contact author has declared that none of the authors has any competing interests.
- **Acknowledgements.** Analysing wave-mean flow interactions and wave activity fluxes, statistical processing was supported by the National Natural Science Foundation of China (Grant No. W2412059); numerical simulations and MUAM adjustment was supported by the Russian Science Foundation (grant #25-47-00122).

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
