# Peer review of "Natural Tropical Oscillations phase impact on stationary and westward travelling planetary waves"

_EGUsphere, 2025_

## Referee Comment (RC1)

Review of **Natural Tropical Oscillations phase impact on stationary and westward travelling planetary waves** by Didenko et al.

This paper presents a thorough investigation of how combinations of ENSO and QBO phases impact planetary waves (both stationary and traveling). They use an idealized middle and upper atmosphere model to investigate and present detailed results of implications on planetary waves 1 and 2, and westward traveling waves of various periods. I found this to be an excellent study that is well motivated, well structured, and well presented. I have a few minor comments that I recommend the authors consider before publication.

General Comments:

1. Introduction: I really enjoyed reading this introduction, it is one of the best I've reviewed in some time. It was informative, well-structured, and comprehensive, and did a good job of motivating the work. Excellent.
2. Results: Your results section presents a lot of information in a very concise manner. I would like a little bit more discussion of the implications of these results. For instance, you motivated the work very well by discussing the relationship of these waves with the polar vortex and SSWs. Based on these results, what can you say regarding the potential impacts of these oscillations on something like the polar vortex?

Specific Comments:

3. Line 56: I would add the formation of cold waves across other regions in addition to east Asia, like North America for instance
4. Line 93: Mention approximate heights of the F2 layer / top of the dataset. Additionally, more details about the vertical and horizontal resolutions of the model would be nice
5. Line 103 – 104: consonant → consistent
6. Line 106: Friedrich et al. (1993); Wallace et al. (1993) → Friedrich et al. (1993) and Wallace et al. (1993)
7. Line 108: "In this work one of the latest versions of the MUAM was used", which version exactly?
8. Line 133: Important advantage → An important advantage

---

## Author Response (AR1)

**Dear Reviewer #1,**

Author and all co-authors would like to thank you for your attention to the publication, high valuation, and useful comments. The corrections and additions made to the manuscript in accordance with Reviewers comments are highlighted. Explanations of the comments are provided below in bold font.

**General Comments:**

1. Introduction: I really enjoyed reading this introduction, it is one of the best I've reviewed in some time. It was informative, well-structured, and comprehensive, and did a good job of motivating the work. Excellent.

**Thank you!**

2. Results: Your results section presents a lot of information in a very concise manner. I would like a little bit more discussion of the implications of these results. For instance, you motivated the work very well by discussing the relationship of these waves with the polar vortex and SSWs. Based on these results, what can you say regarding the potential impacts of these oscillations on something like the polar vortex?

In one of our recent studies, we evaluated the effect of various combinations of QBO and ENSO phases on the SSW formation (Lifar et al., 2024). As you correctly noted, the formation in the SSW is inextricably linked to the intensity of the stratospheric polar vortex. We have found that SSW was observed in 9 out of 10 model runs under El Niño + eQBO in January-February with 4 major SSW. Under El Niño + wQBO SSW was modeled in 8 out of 10 runs with 2 major ones. SSW was modeled in half of the runs with one major during the cold phase of the QBO + eQBO. SSW is not modeled under La Niña + wQBO.

Lifar, V. D., Didenko, K. A., Koval, A. V., and Ermakova, T. S.: Numerical Simulation of QBO and ENSO Phase Effect on the Propagation of Planetary Waves and the Evolvement of Sudden Stratospheric Warming, Atmos. Ocean. Opt., 37, 415–421, doi:10.1134/S1024856024700489, 2024.

Particular thanks for the specific comments. Corrections have been made to the manuscript (L235-238 in Author's track-changes file).

**Specific Comments:**

1. Line 56: I would add the formation of cold waves across other regions in addition to east Asia, like North America for instance

The section and references have been expanded, thank you (L55-68 in Author's track-changes file).

1. Line 93: Mention approximate heights of the F2 layer / top of the dataset. Additionally, more details about the vertical and horizontal resolutions of the model would be nice

The description of the MUAM spatial resolution has been expanded (L100-107 in Author's track-changes file).

1. Line 103 - 104: consonant -> consistent

Revised (L117 in Author's track-changes file).

1. Line 106: Friedrich et al. (1993); Wallace et al. (1993) -> Friedrich et al. (1993) and Wallace et al. (1993)

Thanks, revised (L120 in Author's track-changes file).

1. Line 108: "In this work one of the latest versions of the MUAM was used", which version exactly?

Historically, the model version is not explicitly specified in its name. As indicated in the section you mentioned, the latest version of the MUAM is determined by the parameterizations of atmospheric heating rates caused by the release of latent heat inclusion.

1. Line 133: Important advantage -> An important advantage

Thanks, revised (L157 in Author's track-changes file).

Yours sincerely. K.A. Didenko and co-authors

**Dear Reviewer #2,**

Author and all co-authors would like to thank you for your attention to the manuscript and useful comments. We would also like to express our particular thanks for a constructive discussion of the planetary waves structures. The corrections and additions made to the manuscript in accordance with Reviewers comments are highlighted. Our response is provided below in bold font.

The paper investigates the influence of the ENSO and QBO tropical oscillations on various planetary waves (PWs), namely the stationary PW with zonal wavenumbers 1 and 2 (SPW1, SPW2) and westward travelling quasy 5-, 10- and 7- day PWS. The authors use the MUAM code and consider an ensemble of 10 runs for each of the four combinations of El Niño/La Niña and easterly/westerly QBO. A total of 40 simulations was run.

The authors found that the tropical oscillations can significantly change the amplitudes of the PWs, and detail the specific changes that the different combinations produce. The design of the experiment is interesting and the science results a novel contribution to the literature. However I recommend major revisions to the manuscript before publication for the reasons below.

- The conclusion that the structure of SPW2 amplitudes is basically opposite to SPW1 is not entirely apparent to me. SPW2 under El Niño shows a decrease in amplitudes around the climatological peak, but SPW1 and SPW2 results at La Niña conditions just seem different, not opposite. Please could you clarify which part you mean is the opposite, otherwise please could you take another look at the conclusion

Thank you, clarifications have been added to the Conclusion (L333-338 in Author's track-changes file).

- The conclusion that similar features in are seen in SPW1 and SPW2 amplitude increments during El Niño regardless of QBO phase is not apparent to me. The distribution of positive and negative increments is different between SPW1 and SPW2. Please could you clarify which parts are similar, otherwise please could you take another look at reformulating this conclusion.

**If we understood you correctly, the comment concerns the second point in the Conclusion. This result was obtained in the MLT region.**

- You mention that the 5-day NM amplitude sees a change in flux direction to point from south to north for the eQBO phase, and vice versa for the wQBO phase, regardless of ENSO phase. However I see a decrease in fluxes around the climatological peak during the La Niña/eQBO phase, and an increase for El Niño/wQBO. Please could you clarify which parts of the figures you are drawing your conclusions from, or investigate the conclusion further.

**Clarifications concerning the horizontal transfer area were included in the discussed paragraph of the Conclusion (L366 in Author's track-changes file).**

- Whilst the overall presentation and figures is of high quality, there are a significant improvements required to the writing and grammar before I would recommend publication. I have detailed some below, but ask that the authors check the manuscript carefully and make the relevant sentence structure and grammatical changes.

Thank you for your comments. The text of the publication has been revised.

**Minor comments**

- Please could you make your simulation set-up more clear. It is not clear to me what years you have picked for your simulations and how you have curated your ensemble. If you could perhaps show a table which details some more information about the simulations, e.g., what years are modelled and what stages of ENSO and QBO they are, that would be helpful.

Paragraph 2.1 has been expanded to include a more detailed description of the MUAM simulation set-up.

- L134 - 140. This portion of the paragraph highlights how well MUAM reproduces the structures of PWs, however it is not shown how. I appreciate MUAM PW performance has been discussed in various studies, as you highlight, but perhaps you could at least give the reader an idea of how well MUAM performs in your wavelet analysis.

**Paragraph 2.2 has been expanded.**

- L195 - You mention that the increase of PW amplitudes is attributed to secondary gravity waves by the authors, but then mention other processes such as tides and wave-wave interactions, and not secondary gravity waves. Could you please clarify what you are highlighting here?

Different mechanisms for the formation of amplitude maxima above the region of waveguide interruption are indicated in the specified references, including modulation by gravity waves and planetary ones. Explanations have been added to paragraph 3.1.

- Paragraph starting on L 108. Could you comment on the accuracy of the parameterization and the dependence on ENSO, and the sensitivity of the results to the parameterization? Would the results be much different if a different parameterization was used?

The accuracy of parameterization and the dependence on ENSO used in current work were discussed in detail in Ermakova et al. (2019) (the reference is also presented in the manuscript). Parameterization development and evaluation of their accuracy are out of the scope of this study.

Ermakova, T. S., Aniskina, O. G., Statnaya, I. A., Motsakov, M. A., and Pogoreltsev, A. I.: Simulation of the ENSO influence on the extra-tropical middle atmosphere, Earth Planets Space, 71, 8, doi:10.1186/s40623-019-0987-9, 2019.

- L133 "fidelity reproduction" should probably be "reproduction fidelity"

**Corrected (L157 in Author's track-changes file).**

- L 138 "The accuracy of the simulated by the MUAM PWs and their temporal variability has recently been..." should probably be "The accuracy and temporal variability of the PWs simulated by the MUAM has recently been"

**Corrected (L163-164 in Author's track-changes file).**

- L 145 "reaction" -> "reacting"

- L171 "40 model runs (for 4 combinations of ENSO and QBO, 10 runs each)" - no need to repeat

Thanks, revised (L201-202 in Author's track-changes file).

- L176 - "Figs." -> figures

Thanks, the above remarks have been revised (L208 in Author's track-changes file).

- L194 - "gray areas and arrows in Fig. 1a and 2a." It's not clear how this is relevant. Why is SPW2 wave activity flux mentioned separately? Is SPW1 and SPW2 not covered in the first part of the sentence?

**Clarifications were made in the first paragraph of paragraph 3.1.**

- L195 - ""The reasons for the observed PW amplitude increases related to secondary waves generation are presented in ... " what observed PW amplitude increase do you mean here, the orange region around 60N and 90km in SPW1 and SPW2? Could you please clarify?

**Comments were added to the first paragraph of paragraph 3.1.**

- L209 "Upon that in the area of SPW1 climatic maxima" -> "In the area of the SPW1 climatic maximum"

Thanks, revised (L241 in Author's track-changes file).

- L210 "4 times." -> "A factor of 4"

**Thanks, revised (L242 in Author's track-changes file).**

- L 222 - I don't see a large increase in amplitudes and fluxes in the lower thermosphere only in your Figs 1a and 1c. It looks like there is a 1-10% increase at various regions for all cases. It looks like the strongest increase of amplitude in the LT is around 60N 90 km in La Niña/eQBO. Could you please clarify what region you are highlighting?

We focused on effects up to 120 km in this article, although calculations were performed up to 300 km. However, a detailed review of the mechanisms of PW propagation into the thermosphere remains beyond the scope of this study, so we will focus on the upper atmosphere in the following works. The text of the article has been edited at the paragraph 3.1 and the Conclusion.

- L 225 - It doesn't look like SPW2 has a unique distinction between El Niño and La Niña in the thermosphere in Figs b - e. If you mean the changes in the local max at 60N 120km, which indeed show the behaviour you describe, please could you clarify that in the text.

**Clarifications were made to the text at the paragraph 3.1 and the Conclusion.**

- L 232 "transmitting" -> "transmission"

**Corrected (L267 in Author's track-changes file).**

- L235 "mesosphere – the" -> "mesosphere and the"

Corrected (L270 in Author's track-changes file).

- L 236 ""thermosphere, the mid-latitude" -> "thermosphere and the mid-latitude"

Thanks, the above remarks have been revised (L272 in Author's track-changes file).

Yours sincerely. K.A. Didenko and co-authors

---

## Referee Report (RR1)

A re-review of the article "Natural Tropical Oscillations phase impact on stationary and westward travelling planetary waves" by Kseniia A. Didenko et al.

The authors have done a thorough and high-quality job of finalizing the article, taking into account all comments made in the previous reviews. The changes made have strengthened the scientific value, clarity, and validity of the presented study. The introduction and conclusion have been expanded and revised. A detailed description of the numerical experimental design in MUAM has also been added; this addition is the most significant improvement in terms of scientific rigor and reproducibility. Key findings on the influence of QBO/ENSO phase combinations on the amplitudes and wave activity fluxes of different PWs (SPW1, SPW2, 5-, 10-, 7-day waves), including shifts in latitudinal distribution and amplification/weakening effects, are presented clearly and supported by the results. Thanks to these revisions, the article has reached a high level of quality. This work makes a valuable contribution to understanding the complex interaction of tropical oscillations and planetary waves in the atmosphere and their role in large-scale circulation. I recommend accepting the article for publication in its current form.

---

## Author Response (AR2)

Author and all co-authors would like to thank all Reviewers once again for their attention to the manuscript, which allowed it to be further improved. The clarifications have been made to the text in accordance with the Reviewers minor comments.

**Reviewer #1**

The authors have satisfactorily addressed my comments and the resulting manuscript is much improved. My only remaining comment is that I believe the discussion section could still benefit from a bit more nuance and detail about the broader implications. In the revision, you addressed my comments about the implications of the waves on the polar vortex with the addition of statements like "and therefore affects the strength of the stratospheric polar vortex". While this is an improvement, I think your paper would benefit from a little more detail about (1) what ways the strength of the vortex is affected and (2) why this matters. Other than this, I think the paper should be accepted for publication.

Discussion section have been extended (L379-386 in Author's track-changes file), and references were added (L413-414, 419-420, and 557-559 in Author's track-changes file).

**Reviewer #2**

The authors have done a thorough and high-quality job of finalizing the article, taking into account all comments made in the previous reviews. The changes made have strengthened the scientific value, clarity, and validity of the presented study. The introduction and conclusion have been expanded and revised. A detailed description of the numerical experimental design in MUAM has also been added; this addition is the most significant improvement in terms of scientific rigor and reproducibility. Key findings on the influence of QBO/ENSO phase combinations on the amplitudes and wave activity fluxes of different PWs (SPW1, SPW2, 5-, 10-, 7-day waves), including shifts in latitudinal distribution and amplification/weakening effects, are presented clearly and supported by the results. Thanks to these revisions, the article has reached a high level of quality. This work makes a valuable contribution to understanding the complex interaction of tropical oscillations and planetary waves in the atmosphere and their role in large-scale circulation. I recommend accepting the article for publication in its current form.

**Thank you!**

**Reviewer #3**

I am satisfied with the revisions to the paper introduced in the methods, discussion and conclusions. I recommend the paper be published once the following minor technical clarifications are addressed.

L 106 in tracked changes document "Ion friction, molecular and turbulent viscosity..."

I am assuming these are not part of the radiation unit, and part of a "dynamical unit"? I am not deeply familiar with the MUAM infrastructure, however if they are part of the radiation unit, please could you caveat that "Heating due to ion friction, ... and 3-dimensional ozone distribution are also taken into account"

Clarifications have been added to the text in the specified section (L101-102 in Author's track-changes file).

L 136 in tracked changes: "implicates" -> "activates"?

Thanks, revised (L130 in Author's track-changes file).

L 167 in tracked changes - "Besides ... were founded in EOS MLS"

The meaning here is not clear to me, were the various wave amplitude maxima, and the PW nonlinear interactions simulated by MUAM consistent with EOS MLS? Please could you make the meaning more clear, e.g., change "besides" -> "Additionally", and "founded" -> "found" if so.

Clarifications have been added to the text in the specified section (L159-161 in Author's track-changes file).

Yours sincerely. K.A. Didenko and co-authors